# Microstructure and Mechanical Properties of As-Cast Al-10Ce-3Mg-xZn Alloys

**DOI:** 10.3390/ma17163999

**Published:** 2024-08-12

**Authors:** Haiyang Zhang, Mingdong Wu, Zeyu Li, Daihong Xiao, Yang Huang, Lanping Huang, Wensheng Liu

**Affiliations:** National Key Laboratory of Science and Technology on High-Strength Structural Materials, Central South University, Changsha 410083, China; 223311047@csu.edu.cn (H.Z.); mmingshiddong@csu.edu.cn (M.W.); 16620176815@163.com (Z.L.); youngh@csu.edu.cn (Y.H.); christie@csu.edu.cn (L.H.); liuwensheng@csu.edu.cn (W.L.)

**Keywords:** Al-Ce-Mg alloys, Zn addition, Al_11_Ce_3_ phase, microstructure, mechanical properties

## Abstract

The microstructure and mechanical properties of as-cast Al-10Ce-3Mg-xZn (x = 0, 1, 3, 5 wt.%) alloys were systematically investigated, with a focus on the effect of Zn on the Al_11_Ce_3_ reinforcing phase in the alloy. The results showed that the Al-10Ce-3Mg alloy consists of α-Al, a Chinese-script Al_11_Ce_3_ eutectic phase, and a massive Al_11_Ce_3_ primary phase. With the addition of Zn content, most of the Zn atoms are enriched in the Al_11_Ce_3_ phase to form the acicular-like Al_2_CeZn_2_ phase within the Al_11_Ce_3_ phase. Increasing the Zn content can increase the volume fraction of the Al_11_Ce_3_ phase. Compared to the alloy without Zn addition, the microhardness and elastic modulus of the Al_2_CeZn_2_-reinforced Al_11_Ce_3_ phase in the alloy with 5 wt.% Zn increased by 18.9% and 9.0%, respectively. Moreover, the room-temperature mechanical properties of Al-10Ce-3Mg alloys were significantly improved due to the addition of Zn element. The alloy containing 5 wt.% Zn had the best tensile properties with an ultimate tensile strength of 210 MPa and a yield strength of 171MPa, which were 21% and 77% higher than those of the alloy without Zn, respectively. The alloy containing 5 wt.% Zn demonstrated a superior retention ratio of tensile strength at 200–300 °C, indicating that the alloy has excellent heat resistance. The improvement in the mechanical properties is primarily attributed to second-phase strengthening and solid solution strengthening.

## 1. Introduction

Aluminum alloys have been commonly used in aerospace and automotive industries due to their excellent castability, superior mechanical properties, and low cost [1,2,3]. However, most aluminum alloys exhibit the characteristic of “light weight and high strength” only at room temperature, and it is difficult to sustain this advantage at 200–300 °C [4]. For existing precipitation-hardening aluminum alloys, such as Al-Cu and Al-Zn-Mg series alloys, the reinforcing precipitation phases in the alloys tend to coarsen or dissolve at elevated temperatures above 200 °C [5,6,7]. As a result, the high-temperature mechanical properties of aluminum alloys retain only a small fraction of the room-temperature mechanical properties. Therefore, there is an immediate requirement for a different aluminum alloy design paradigm to design a heat-resistant aluminum alloy that can be utilized at temperatures ranging from 200 to 300 °C.

Recently, Al-Ce alloys with the rare earth element Ce as the main element were reported to have superior castability and heat resistance due to the high thermal stability of the Al_11_Ce_3_ phase [8,9,10]. The significant differences in the crystal structure and atomic radius of the two elements are responsible for the low solubility of Ce in Al near the eutectic point (642 °C), with the upper limit of Ce solubility in Al being below 0.05 wt.% [11,12,13]. As a result, the alloy tends to form a large number of Al_11_Ce_3_ phases during solidification [14,15]. The excellent heat resistance of the alloy is due to the high thermal stability of the Al_11_Ce_3_ phase, which has a melting point of approximately 1253 °C. Czerwinski et al. [15] conducted experiments on Al-Ce binary alloys to compare the properties with those of the commonly used A380 alloy (Al-Si-Cu system). The Al-Ce binary alloy was found to maintain the better strength retention ratio within the temperature range of 200–300 °C.

However, the compositional design of binary Al-Ce eutectic alloys is simpler than that of the heat-resistant aluminum alloys currently used in industry. The alloy matrix has a lower concentration of solute, resulting in a significant difference in strength between the matrix and the intermetallic phase [16]. When the stress is applied, a large number of primary intermetallic compounds lead to the stress concentration and microcracking, resulting in poor mechanical properties at room temperature [17]. To enhance the mechanical properties of alloys, the techniques such as optimizing the alloy composition [18,19], refining the casting process [20,21], and implementing deformation treatment [22,23] have been used. Among them, optimizing the composition of Al-Ce alloys can be considered as the simpler and effective method. To date, adding specific elements to improve the solid solution strengthening, improve the precipitation strengthening effect, or refine the primary Al_11_Ce_3_ phase is the main way to improve the room temperature properties of the alloy. For instance, by adding Mg to Al-Ce alloys, Wang et al. [24] found that Mg primarily provides solid solution strengthening in Al-Ce alloys and can significantly enhance the room-temperature mechanical properties of Al-Ce eutectic alloys. Wang et al. [25] discovered that the addition of Y to the Al-Ce alloy resulted in the formation of a new ternary phase, Al_4_(Ce, Y), which was the form of submicron fibers. Then, the strengthening effect of the Al_4_(Ce, Y) phase was investigated by experiments and simulations, and the results demonstrated that the Al-8Ce-3Y alloy broke through the strength limit of Al-Ce casting alloys. Ye et al. [19] found that the addition of Zr refined the primary Al_11_Ce_3_ phase of the Al-15Ce alloys. Compared to the Zr unalloyed alloy, the alloy with Zr addition exhibited an increase in ultimate tensile strength (UTS) from 116.8 MPa to 181.2 MPa, yield strength (YS) from 74.5 MPa to 145.1 MPa, and elongation (EL) from 1.4% to 3.4%.

Based on the above studies, it is found that the impact of adding a single element on the properties of Al-Ce alloys remains limited, and the compositional design is relatively straightforward. Therefore, additional alloying elements must be incorporated to optimize the alloy composition. For conventional aluminum alloys, the element Zn enhances alloy properties through solid solution strengthening, similar to the element Mg [26,27]. However, it has been found that the addition of either Zn or Mg alone has limited impact on improving the overall mechanical properties of the alloy [28]. In Al-Zn-Mg-(Cu) alloys, the main alloying elements Zn and Mg can be precipitated to form many fine and dense T or η phases after appropriate heat treatments, showing remarkably improved strength. Inspired by this, the combined additions of Zn and Mg in Al-Ce alloys can result in a stronger strengthening effect. In addition, since the atomic radius of Zn is smaller than that of Al, it may replace the position of Al atoms to form a new phase. Khan et al. [29] found the new (Al, Zn)_3_Zr phase in the Al-Zn-Mg-Cu alloy. Therefore, the addition of Zn in Al-Ce alloys may lead to the entry of Zn into the Al_11_Ce_3_ phase to enhance the second-phase strengthening effect and improve the properties of the alloy. 

At present, there are limited studies on the effect of Zn content on Al-Ce alloys, and it remains to be investigated whether the combined additions of Zn and Mg in Al-Ce alloys can result in a stronger strengthening effect. Thus, in order to further optimize the alloy compositions of Al-Ce alloys, it would be an interesting research direction to investigate the impact of varying Zn contents on the microstructure and mechanical properties of Al-Ce-Mg alloys. Moreover, the reserves and production of rare earth element Ce are abundant worldwide [30,31]. Improving the performance of Al-Ce alloys and broadening their scope of application can increase the supply and price of Ce, effectively alleviating the imbalance in the application of rare earth elements [32,33]. The further wide application of lightweight aluminum alloys also makes an important contribution to energy conservation and improving the living environment.

In this study, the effects of different Zn contents on the microstructure and mechanical properties of as-cast Al-Ce-Mg alloys were investigated, and the related mechanisms are elucidated in this paper. This study provides a new strategy to improve the high-temperature properties of Al-Ce alloys, which further improves their wide application.

## 2. Materials and Methods 

Based on previous calculations of the Al-Ce binary phase diagram [34,35] and the Al-Ce-Mg ternary phase diagram [36], Al-10Ce-3Mg alloys with different Zn contents were designed. The alloy ingots were prepared using pure Al (99.99 wt.%), pure Mg (99.99 wt.%), pure Zn (99.99 wt.%), and intermediate alloys including Al-30 wt.% Ce, Al-5 wt.% Zr, and Al-10 wt.% Y. After completely melting pure Al in a graphite crucible at 850 °C using a resistance furnace, a predetermined number of intermediate alloys were wrapped in aluminum foil and added to the aluminum melt. When the intermediate alloys were completely melted, the melt was stirred thoroughly to ensure homogeneity. Then, the entire melt was degassed using argon and refined with C_2_Cl_6_. Finally, the melt was maintained at a temperature of 800 °C for 15 min before being cast into a steel mold with a dimension of φ120 × 200 mm. The chemical composition of the alloy specimens was analyzed by Inductively Coupled Plasma Atomic Emission Spectrometry (ICP-AES, Spectro Blue SOP, SPECTRO, Clive, Germany), and the results are shown in Table 1.

The samples taken from the same area of the ingots were ground and polished. The polished samples were immersed in NaOH solution (20% NaOH, 80% distilled water by weight) for 5 min and then rinsed in deionized water to reveal the 3D morphologies of intermetallic phases. The alloys underwent physical phase analysis using the X-ray diffractometer (XRD, D8 Advance, Bruker, Berlin, Germany) with a scanning speed of 5°/min and a scanning range of 10–90°. The microstructure and morphology of the alloys were analyzed by scanning electron microscopy (SEM, MIRA4 LMH, TESCAN, Brno, Czech Republic), and the chemical element content of the phases in the alloy was quantified through energy dispersive spectroscopy (EDS, Ultim Max 50, Oxford instrument, Oxford, UK). The elemental distribution in the alloys was determined using electron probe X-ray microanalysis (EPMA, JXA-8230, JEOL, Akishima City, Japan). The detailed microstructure of the phase was further characterized using transmission electron microscopy (TEM, Talos F200X, Thermo Fisher Scientific, Waltham, MA, USA) equipped with an EDS (Super-X, Thermo Fisher Scientific, USA) at 200 kV.

Nanotips for atom-probe tomography (APT) analysis were prepared in a dual-beam focused ion beam scanning electron microscopy (FIB-SEM, HELIOS 5UC, Thermo Fisher Scientific, USA) using a standard lift-out technique. Moreover, the nanotips were prepared from random positions on the primary phase in the alloy. APT experiments were performed using a Local Electrode Atom Probe (LEAP4000X Si, CAMECA, Gennevilliers, France) with a UV laser pulse repetition rate of 200 kHz, at a specimen temperature of 20 K, a target evaporation rate of 0.5%, and a pulsing laser energy of 40 pJ under a high vacuum of 5.0 × 10−11 Torr. The CAMECA Visualization and Analysis Software (IVAS 3.8.4) package was used to reconstruct, visualize, and analyze the APT datasets.

To obtain mechanical information on the intermetallic phases, the specimens were subjected to nanoindentation experiments using a nanoindentation tester (MCT + UNHT, CSM, Geneva, Switzerland). Each sample underwent five tests to determine the mean value. The software of the instrument provided measurements for microhardness (*H*) and elastic modulus (*E*), and fracture toughness (*K*_1*C*_) was calculated by the following equation [37]:(1)$$K_{1C}=0.016{\left(\frac{E}{H}\right)}^{0.5}(\frac{F}{c^{1.5}})$$
where *F* is the indentation force (500 mN), and *c* is the length of the crack emanating from the center of the indentation. The Vickers hardness of the alloys was measured using a Vickers hardness tester (200HV-5, Huayin, Changsha, China) with a load of 5 kg and a dwell time of 15 s. The average of five points was calculated. The cylindrical tensile specimens, measuring 60 mm in length and 5 mm in diameter, were obtained through lathe machining. These specimens were then subjected to uniaxial tensile testing using a mechanical testing machine (Instron 3609, Instron, Norwood, MA, USA). The tests were conducted at room temperature. The alloys were each tested three times at a tensile rate of 2 mm/min, and the results were averaged.

In order to study the heat resistance of the alloys, the high-temperature tensile tests were carried out at 200 °C, 260 °C, and 300 °C with a Zwick/Roell Z100 testing machine (Zwick Roell, Ulm, Germany) at a strain rate of 1 mm/min. The dog-bone-shaped sample had a length of 32 mm, a width of 3.5 mm, and a thickness of 1.5 mm. Each sample was tested three times, and the results were averaged. In order to study the fracture behavior of the alloys, the microstructure near the fracture was analyzed using SEM.

## 3. Results 

### 3.1. As-Cast Microstructure Characterization

The XRD patterns of the studied alloys are shown in Figure 1. It can be seen that α-Al and the Al_11_Ce_3_ phase are present in all alloys. The intensity of the Al_11_Ce_3_ peak increases with the addition of Zn, as shown in the red dotted line in Figure 1. This indicates that the volume of the Al_11_Ce_3_ phase increases with higher Zn content. Furthermore, new peaks are present in the spectrum of alloy 4. By comparison with the standard card, the phase can be identified as Al_2_CeZn_2_. This demonstrates that following the introduction of 5 wt.% Zn, the new ternary Al_2_CeZn_2_ phase appears in the alloy.

Figure 2 displays the SEM micrographs of as-cast Al-10Ce-3Mg alloys with varying Zn contents. All alloys comprise α-Al, a eutectic phase, and a primary phase, which is a typical feature of peritectic alloys (Figure 2a). The eutectic colonies are dendritically distributed, and there are inter-branch channels between the eutectic groups (Figure 2b). The single eutectic colony shows a Chinese-script eutectic morphology and is formed by alternating α-Al and intermetallic flakes (Figure 2c). The massive primary phase is in the form of a block with a length of several tens of microns, which is surrounded by fine eutectic clusters (Figure 2d). These microstructural observations align with previous reports on cast Al-Ce alloys [36].

The 3D morphologies of the intermetallic phases of Al-10Ce-3Mg alloys with different Zn contents are shown in Figure 3. For imaging in the secondary electron mode, the intermetallic phase shows a bright contrast, while α-Al appears as dark gray. In the intermetallic phases, the primary phases are characterized by columns with sharp edges and complex cross sections (Figure 3(a1,b1,c1)), and the eutectic phases are long and curved sheets with cross-sections of Chinese-script morphologies (Figure 3(a2,b2,c2)). Moreover, all intermetallic phases are embedded in the α-Al matrix. The chemical composition of the points in Figure 3 was determined by EDS, and the composition of each phase obtained is shown in Table 2. The phase is identified as the Al_11_Ce_3_ phase based on the EDS results of the marked point in Figure 3(a1). The EDS point analysis in Figure 3(a2) indicates that a small amount of Mg is solidly dissolved in the eutectic Al_11_Ce_3_ phase. With the addition of Zn, the element Zn is found to enter into the intermetallic phases, and the Zn content in the α-Al matrix and intermetallic phase increases. Yang et al. [38] discovered that Zn has a high solubility in Al_11_Ce_3_ and can create ternary phases, including Al_11−x_Ce_3_Zn_x_ and Al_2_CeZn_2_. Combined with the EDS results of the points in Figure 3(b1,c1), it is judged that these phases are Al_11−x_Ce_3_Zn_x_ phases. Therefore, the microstructures of alloy 2 and alloy 3, respectively, both consist of an Al_11−x_Ce_3_Zn_x_ primary phase and an AlCeMgZn eutectic phase. The EDS analysis reveals that the phase in alloy 4 is Al_2_CeZn_2_, as shown in Figure 3(d1). The microstructure of the alloy consists of an Al_2_CeZn_2_ primary phase and an AlCeMgZn eutectic phase.

In order to further determine the effect of Zn addition on the intermetallic phases, EPMA tests were carried out on alloy 1 and alloy 4, respectively. Figure 4 displays the elemental distribution of Al, Ce, Mg, and Zn in the alloys. As shown in Figure 4a, the Ce element in alloy 1 is concentrated in the Al_11_Ce_3_ primary phase and Al_11_Ce_3_ eutectic phase, and the Mg element is primarily dissolved in the α-Al matrix. However, the distribution of the other elements in alloy 4 remains unchanged, with the Zn element primarily concentrated in the Al_11_Ce_3_ phase (Figure 4b). Therefore, based on the distribution of the Ce and Zn elements, it can be inferred that the addition of 5 wt.% Zn to the Al-10Ce-3Mg alloy results in the reaction of the Al_11_Ce_3_ phase with Zn, forming a new Al_2_CeZn_2_ phase.

In order to further investigate the presence mode of elemental Zn in the Al_11_Ce_3_ phase, the massive primary phases in Al-10Ce-3Mg cast alloys containing 0 wt.% Zn and 5 wt.% Zn were characterized using High-angle annular dark-field (HAADF)-STEM and EDS compositional mapping, respectively (Figure 5). Figure 5a–c show the TEM images of the massive primary phase of alloy 1, while Figure 5d–g show the TEM images of the massive primary phase of alloy 4 and the corresponding EDS mapping results. The α-Al matrix is represented by the grey area in Figure 5a, and the other areas correspond to the massive primary phase. Further magnification shows that there is no atomic segregation in the massive primary phase (Figure 5b), which is identified as the pure Al_11_Ce_3_ phase by the associated selected area electron diffraction (SAED) map (Figure 5c). However, Figure 5e presents a further magnified HAADF-STEM image of the massive primary phase in Figure 5d, and there is a large amount of needle-like segregation with white contrast. Moreover, the white phase is acicular, with a certain angle between adjacent acicular phases, and all the phases are uniformly dispersed in the massive primary phase. The elements in the region of Figure 5e were analyzed by EDS mapping. An observation of the green arrowed region in Figure 5e reveals that the green peaks are Zn elemental peaks, proving that there is a significant bias towards the Zn element. At the same time, it is found that the area where the Zn element is biased coincides exactly with the area where the white phase is present (Figure 5g). Combining the results of XRD and SEM, the white acicular phase was identified as an Al_2_CeZn_2_ phase. The result demonstrated that Zn was biased in the Al_11_Ce_3_ phase and led to the formation of the new Al_2_CeZn_2_ phase within the Al_11_Ce_3_ phase. Analyzing the associated SAED plots (Figure 5f) and comparing them to Figure 5c, it can be seen that there is no significant change in the diffraction spots and the Al_2_CeZn_2_ phase is co-lattice with the Al_11_Ce_3_ phase. 

The microstructure of the primary phase in alloy 4 was further investigated using atom probe tomography (APT). Figure 6a shows the 3D-APT reconstruction of the nanotip obtained from the primary phase, where it is evident that Zn-rich precipitates are contained in the Al_11_Ce_3_ primary phase. Figure 6c illustrates the interface between the primary phase and the Zn-rich precipitate in the alloy. The content of Zn initially increases and then decreases, while the content of Al decreases initially and then increases. However, the content of Ce remains unchanged. This demonstrates that some of the Al in the Al_11_Ce_3_ phase is replaced by Zn, resulting in the transformation of the Al_11_Ce_3_ phase into the Al_2_CeZn_2_ phase. The primary phase consists of the Al_11_Ce_3_ and Al_2_CeZn_2_ phases.

### 3.2. Mechanical Properties

Nanoindentation experiments were conducted on the massive primary phase to examine the impact of the diffusely distributed Al_2_CeZn_2_ phase on the Al_11_Ce_3_ phase. Figure 7a shows the nanoindentation curves of different alloys. The results show that the microhardness (H) and elastic modulus (E) of the massive primary phase in alloy 4 are 8.3 ± 0.1 GPa and 128.4 ± 4.2 GPa, respectively. These values are higher than those of the massive primary phase in alloy 1, which are 6.9 ± 0.2 GPa and 115.9 ± 2.0 GPa. The values of H and E of the primary phase in alloy 2 are 7.6 ± 0.3 GPa and 117.4 ± 3.5 GPa, respectively, and those of the primary phase in alloy 3 are 8.1 ± 0.2 GPa and 126.4 ± 3.6 GPa, respectively. The residual Berkovich impressions are shown in Figure 7b,c. The length of c was measured as 24.71 μm and 23.46 μm for alloy 1 and alloy 4, respectively. By substituting the obtained indentation data into Formula 1, the K_1C_ was calculated to be 0.26 ± 0.01 MPa·m^1/2^ and 0.28 ± 0.01 MPa·m^1/2^ for the primary phases of alloy 1 and alloy 4, respectively. This indicates that the massive primary phases of both alloys are extremely brittle. In contrast, the primary phases of alloy 4 exhibit higher hardness and elastic modulus, indicating that the Al_2_CeZn_2_ phase enhances the strength of the Al_11_Ce_3_ phase and, subsequently, the overall strength of the alloy through second phase strengthening. In addition, the fracture toughness of the primary phase in alloy 4 is greater than alloy 1, suggesting that the Al_2_CeZn_2_ phase improves the ability of the Al_11_Ce_3_ phase to resist a brittle fracture.

Figure 8 displays the hardness values of the Al-10Ce-3Mg alloys with various Zn additions. The data show that increased Zn content leads to an increase in the hardness. Alloy 4 has a maximum hardness of 73.4 ± 0.7 HV, which is an increase of 28 ± 3% compared to the hardness of alloy 1 (57.0 ± 0.8 HV). This is because some of the Zn enters the intermetallic phase to increase its hardness, while the rest is solidly dissolved in α-Al to enhance the hardness of the matrix.

Figure 9a illustrates the impact of Zn content on the mechanical properties of the alloy at room temperature. It can be clearly seen that an increase in the Zn content leads to an increase in the ultimate tensile strength (UTS) and yield strength (YS), and a decrease in elongation (EL). The UTS, YS, and EL of alloy 1 were 173 ± 4 MPa, 97 ± 2 MPa and 2.5 ± 0.2%, respectively. When the Zn concentration reached 5 wt.%, both UTS and YS reached the maximum value of 210 ± 3 MPa and 171 ± 2 MPa, respectively. These values were 21 ± 2% and 77 ± 5% larger than those of alloy 1, while EL decreased by 60 ± 6% to 1.0 ± 0.1%. The experimental results demonstrate that the addition of Zn to the Al-10Ce-3Mg alloys can considerably enhance their room-temperature tensile strength. The decrease in plasticity values of alloy 4 may be attributed to the increase in the number and volume of Al_11_Ce_3_ phases.

In order to evaluate the effect of Zn content on the high-temperature mechanical properties of the alloys, tensile tests were carried out at several high temperatures (200, 260, and 300 °C). As seen in Figure 9b–d, an increase in Zn content caused an enhancement in alloy strength at all test temperatures. Alloy 4 exhibits the highest UTS, with values of 142 ± 2 MPa at 200 °C, 125 ± 5 MPa at 260 °C, and 72 ± 5 MPa at 300 °C. As the test temperature is increased, the serrated phenomenon observed in the stress–strain curve becomes more pronounced. The serration behavior is primarily attributed to the interaction between the intermetallic phases and movable dislocations at elevated temperatures [39,40]. It should be noted that an increase in the test temperature resulted in a reduction in the strength of the alloys. However, the strength retention ratios of the alloy containing 5 wt.% Zn at 200 ° C, 260 ° C, and 300 ° C were 66 ± 2%, 58 ± 2%, and 34 ± 3%, respectively. The strength retention ratio was calculated as the UTS measured at the test temperature divided through the UTS measured at room temperature. Compared with the conventional heat-resistant aluminum alloy [41], the alloy in this study demonstrated a superior retention ratio of tensile strength at the same temperature. This indicates that the alloys with different Zn contents have superior heat resistance.

The room temperature fracture morphologies of the as-cast Al-10Ce-3Mg alloys with various Zn contents were characterized by SEM (Figure 10(a1,b1,c1,d1)). The tensile fracture of the alloys had distinct cleavage facets, which were intermixed with each other. This indicates that the fracture morphology of the alloy exhibited typical brittle fracture characteristics. In addition, there were cracks distributed around the cleavage facets (Figure 10(a1)). The disintegrated surfaces were primarily caused by brittle and coarse Al_11_Ce_3_ intermetallic compounds that fracture due to stress concentration under tensile stress. It can be seen that the percentage of disintegrated surfaces increases with increasing Zn content, while the length of the cracks remains essentially unchanged (Figure 10(d1)). The main reason for the increase in the dissociation surface is the higher percentage of the Al_11_Ce_3_ phase in the alloy resulting from the increased Zn content. The higher amount of the Al_11_Ce_3_ phase increased the strength of the alloy while at the same time increasing the risk of fracture, resulting in a significant reduction in plasticity. The fracture pattern of the alloys tended to be more towards brittle fracture. In general, the fracture morphology remains consistent with the tensile test data presented in Figure 9a.

The longitudinal sectional area near the fracture surface was characterized in order to comprehend the fracture behavior of the alloy, as shown in Figure 10(a2,b2,c2,d2). Microcracks were observed in the Al_11_Ce_3_ phases closest to the fracture (Figure 10(a2)), while the Al_11_Ce_3_ primary phases with smooth cross-sections were observed at the fracture. These phenomena further demonstrate that the Al_11_Ce_3_ phase under tensile stresses is the origin of crack initiation. After the Al_11_Ce_3_ phase fractures, the cracks were interconnected, leading to alloy fracture.

## 4. Discussion

### 4.1. Effect of Zn Addition on Microstructure

The effect of Zn on the microstructure of the Al-10Ce-3Mg alloy is mainly reflected by the fact that the addition of Zn increases the volume fraction of the Al_11_Ce_3_ phase and forms a uniformly distributed acicular Al_2_CeZn_2_ phase in the Al_11_Ce_3_ phase. The results of Figure 1 show that the volume fraction of intermetallic phase in the alloys increases with the addition of Zn content. From the statistical results in Figure 2, the volume fraction of the Al_11_Ce_3_ phase is 11.4%, 13.2%, 14.6%, and 16.9% for alloy 1, alloy 2, alloy 3, and alloy 4, respectively. This is mainly because the addition of alloying elements to eutectic Al-Ce alloys leads to a shift in the eutectic composition point towards the left. This resulted in the generation of the primary phase Al_11_Ce_3_ in eutectic Al-10Ce alloys [36]. Meanwhile, the Zn element is able to reduce the diffusion and migration rate of atoms in the alloy and lower the surface energy of solidification [27]. Consequently, upon solidification of the alloy, the Al_11_Ce_3_ phases precipitate and grow more easily into massive primary phases, resulting in an increase in both the volume and amount of the Al_11_Ce_3_ phases in the alloy. Furthermore, as illustrated in Table 2, the addition of the Zn element causes part of the Zn element to be solidly dissolved in the α-Al matrix, which enhances the solid solution strengthening effect of the alloy. A portion of the Zn element dissolves into the Al_11_Ce_3_ phase, causing it to gradually transform into the Al_11−x_Ce_3_Zn_x_ phase. When the concentration of Zn reaches a certain level, diffusely distributed acicular Al_2_CeZn_2_ phases are formed in the Al_11_Ce_3_ phase. The results are consistent with the XRD curves. The elemental Zn has a large solubility in the Al_11_Ce_3_ phase (maximum solubility of 17.9 and 12.2 at.% at 600 °C and 450 °C, respectively) [38]. As the amount of Zn increases, Zn atoms take the place of Al atoms. Based on the isothermal section diagram of the Al-Ce-Zn phase diagram, it is evident that the Al_11_Ce_3_ phase transforms into the Al_11−x_Ce_3_Zn_x_ phase, which ultimately results in the formation of a new Al_2_CeZn_2_ phase.

### 4.2. Effect of Zn Addition on Strength

The inclusion of Zn in the Al-10Ce-3Mg alloys leads to a significant increase in hardness and improves mechanical properties. The strengthening mechanism of alloys containing particle strengthening phases typically involves solid solution strengthening, fine grain strengthening, precipitation strengthening, second phase strengthening, and deformation strengthening [22,24]. Since the as-cast Al-10Ce-3Mg alloys do not undergo heat treatment or plastic deformation, precipitation strengthening and deformation strengthening are not applicable. The grains of the as-cast Al-Ce alloys are relatively large, and the majority of the intermetallic compounds are embedded within the grains, so the fine grain strengthening effect is not obvious. The strengthening mechanism of the alloy is achieved by the second-phase strengthening and solid solution strengthening. Assuming that the various strengthening mechanisms act independently, their effects are considered to be additive, and the total strength of the current alloy can be expressed as follows [42,43]:(2)$$\sigma_{ys}=\sigma_{0}+\sigma_{sp}+\sigma_{s}$$
where *σ*_0_ is the Peierls or friction stress, *σ_sp_* is the second-phase strengthening, and *σ_s_* is the solid solution strengthening.

The contribution of the Al_11_Ce_3_ phase to the strengthening of the second phase in Al-10Ce-3Mg alloys is discussed according to Brown’s method with reference to previous studies [44,45]:(3)$$\sigma_{sp}=4\phi \gamma G_{Al}f_{Al11Ce3}\varepsilon^{*}$$
where *γ* = 1/2(1 − *ν*) is the accommodation factor, *ν* is the Poisson’s ratio of (0.345) Al, *G_Al_* is the shear modulus of the α-Al matrix (25.4 GPa at 25 °C), *f_Al_*_11*Ce*3_ is the volume fraction of the Al_11_Ce_3_ phase, *ε** is the unrelaxed plastic strain around 0.002 for Al, and *ϕ* is obtained by the following equation [46].
(4)$$\phi =\frac{G_{Al11Ce3}}{G_{Al11Ce3}-\gamma (G_{Al11Ce3}-G_{Al})}$$

*G_Al_*_11*Ce*3_ is obtained by the following equation:(5)$$G_{Al11Ce3}=\frac{\;E_{Al11Ce3}}{2(1+\nu )}$$
where *G_Al_*_11*Ce*3_ is the shear modulus for the Al_11_Ce_3_ phase, *G_Al_* is the shear modulus for the Al phase, and *E_Al_*_11*Ce*3_ is the elastic modulus for the Al_11_Ce_3_ phase. 

The results of the nanoindentation test indicate that the addition of Zn results in an increase in the *E* of the Al_11_Ce_3_ phase in the alloy (Figure 7a). The reason is that the fine acicular Al_2_CeZn_2_ phase is uniformly and diffusely distributed in the Al_11_Ce_3_ phase (Figure 5), which plays a certain role in strengthening the Al_11_Ce_3_ phase. Based on Formulas 4 and 5, it can be observed that when the value of *G_Al_* remains constant, there is a positive correlation between *ϕ* and *G_Al_*_11*Ce*3_, and a positive correlation between *E_Al_*_11*Ce*3_ and *G_Al_*_11*Ce*3_. Thus, an increase in *E_Al_*_11*Ce*3_ results in a corresponding increase in *G_Al_*_11*Ce*3_, which subsequently leads to an increase in the value of *ϕ*. Calculations from Formula 4 show that the values of *ϕ* for each of the four alloys are 1.45, 1.47, 1.53, and 1.55, respectively. In addition, the microstructure illustrates that an increase in the Zn element results in a higher volume fraction of the Al_11_Ce_3_ phase, which, in turn, leads to an increase in *f_Al_*_11*Ce*3_. From the statistical results obtained through SEM images, the volume fraction of the Al_11_Ce_3_ phase is 11.4%, 13.2%, 14.6%, and 16.9% for alloy 1, alloy 2, alloy 3, and alloy 4, respectively. In combination with the above analysis, the second-phase strengthening is primarily influenced by the elastic modulus of the Al_11_Ce_3_ phase and the volume fraction of the Al_11_Ce_3_ phase. It can be seen from Formula 3 that the *σ_sp_* of alloy 4 is about 17, 10, and 7 MPa higher than that of alloy 1, alloy 2, and alloy 3, respectively. The increase in Zn content enhances the second-phase strengthening effect in the alloy. When the alloy is externally stressed, the large Al_11_Ce_3_ primary phases cause entanglement of dislocations and concentration of stress. As stress increases, the Al_11_Ce_3_ primary phases break up and become the source of cracks (Figure 10), ultimately causing the alloy to fracture. The inclusion of Zn in the alloy increases the elastic modulus of the Al_11_Ce_3_ phase, resulting in an increase in the UTS of the alloy. However, the increase in the volume fraction of the Al_11_Ce_3_ phase also leads to an increase in crack sources, resulting in a decrease in the EL of the alloy.

It was discovered that a significant quantity of Zn atoms can dissolve in the Al lattice across a broad temperature range. However, due to the larger atomic radius of Zn compared to Al, it increases the lattice spacing of α-Al, resulting in lattice distortion. This will effectively impede the movement of dislocations, leading to solid solution strengthening of the alloy. The strengthening effect of the solid solution is directly proportional to the concentration of atoms within it. To calculate solid solution strengthening, use the following equation [47,48]:(6)$$\sigma_{s}={\;k}_{j}C_{j}^{2/3}$$
where *C_j_* is the concentration of Zn atoms in solid solution and *k_j_* = 3 MPa/(wt.%)^2/3^ is the corresponding scaling factor [48]. With the increase in Zn content, the *σ_s_* of the four alloys is 0, 3, 6, and 9 MPa.

As the Zn content increases, more Zn solid solution atoms enter the α-Al matrix (Figure 3). Therefore, the stronger the lattice distortion caused by Zn atoms, the more obvious the solid solution strengthening effect, resulting in a higher hardness and strength of the alloy. However, when the alloy is solidified, a part of Zn will be enriched in the Al_11_Ce_3_ intermediate phase to form an Al_2_CeZn_2_ phase. A proportion of Zn atoms is consumed, which leads to deviation between the theoretical value and the experimental data.

## 5. Conclusions

The effect of adding different Zn contents on the microstructure and mechanical properties of the Al-10Ce-3Mg cast alloy was systematically investigated in this work. The following conclusions could be drawn:

(1)The as-cast Al-10Ce-3Mg-xZn alloys consist of α-Al, a Chinese-script eutectic phase, and a bulk primary phase. The volume fraction of the Al_11_Ce_3_ phase in the alloy increases as the Zn content increases, and the majority of the Zn element is concentrated in the Al_11_Ce_3_ phase; when the Zn addition is increased to 5 wt.%, the regularly distributed acicular Al_2_CeZn_2_ phase is formed within the Al_11_Ce_3_ phase.(2)The alloy with 5 wt.% Zn exhibited a significant increase in microhardness, elastic modulus, and fracture toughness of the Al_11_Ce_3_ phase compared to the alloy without Zn addition; this indicates that the increase in the Zn content significantly enhances the second-phase strengthening effect of the alloy.(3)The addition of Zn can significantly enhance the hardness and mechanical properties of Al-10Ce-3Mg alloys through the combined effect of solid solution strengthening and second-phase strengthening. The UTS and YS of the Al-10Ce-3Mg alloy increase with increasing Zn content at room temperature, while the EL gradually decreases. The decrease in EL is due to the increase in the volume fraction of the Al_11_Ce_3_ phase as a source of fracture; the alloy with 5 wt.% Zn exhibits the highest UTS at all test temperatures, indicating that the alloy has superior heat resistance.

## Figures and Tables

**Figure 1 materials-17-03999-f001:**
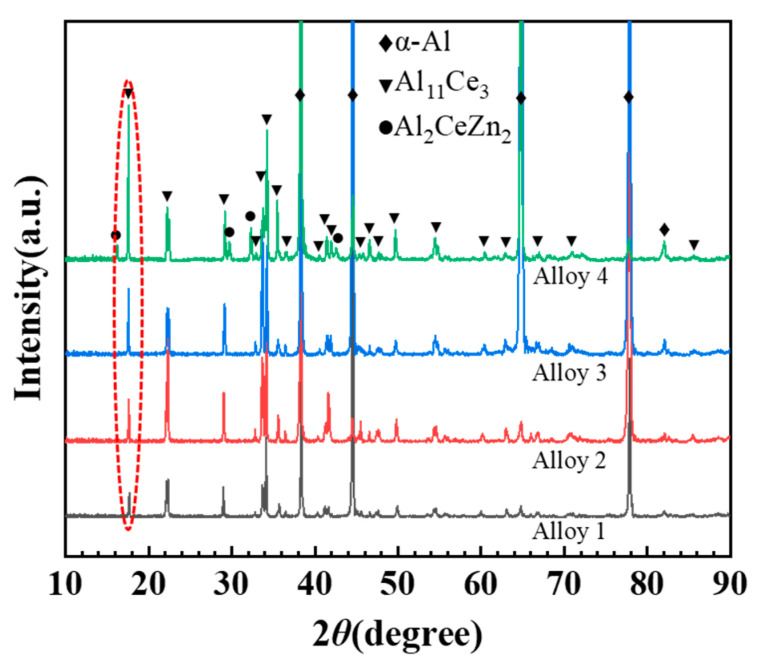
XRD patterns of the as-cast Al-10Ce-3Mg alloys with various Zn concentrations. The red dotted circle indicates that the intensity of the Al_11_Ce_3_ peak increases with increasing Zn content.

**Figure 2 materials-17-03999-f002:**
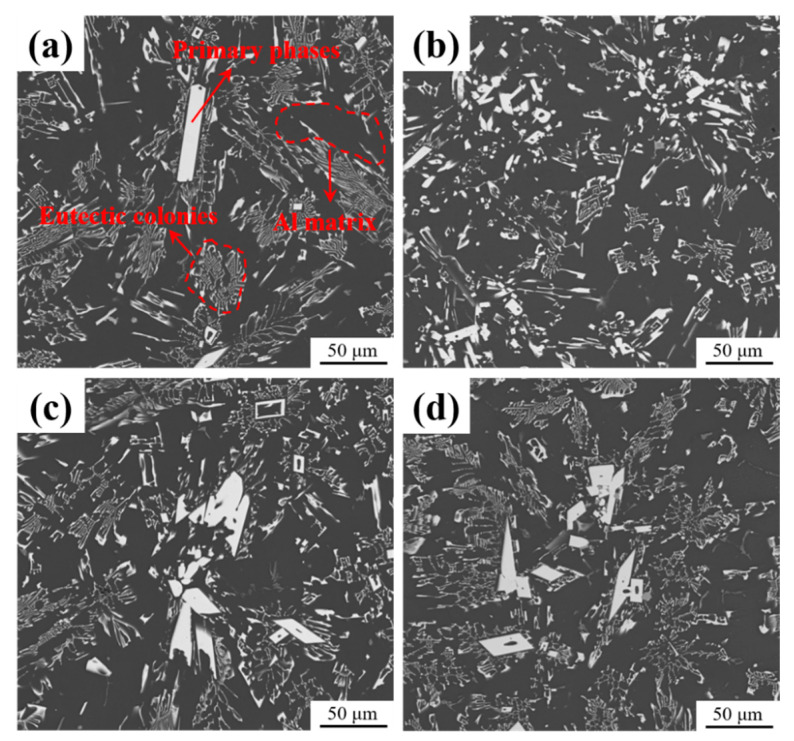
Backscattered SEM micrographs of the as-cast Al-10Ce-3Mg alloys with various Zn concentrations: (**a**) alloy 1, (**b**) alloy 2, (**c**) alloy 3, and (**d**) alloy 4.

**Figure 3 materials-17-03999-f003:**
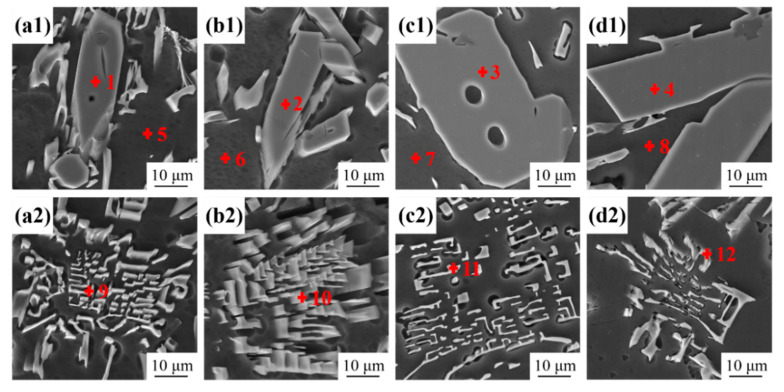
SEM micrographs of the 3D morphologies of intermetallic phases revealed by deep-etching in as-cast Al–10Ce-3Mg alloys with various Zn concentrations: (**a1**,**a2**) alloy 1, (**b1**,**b2**) alloy 2, (**c1**,**c2**) alloy 3, and (**d1**,**d2**) alloy 4.

**Figure 4 materials-17-03999-f004:**
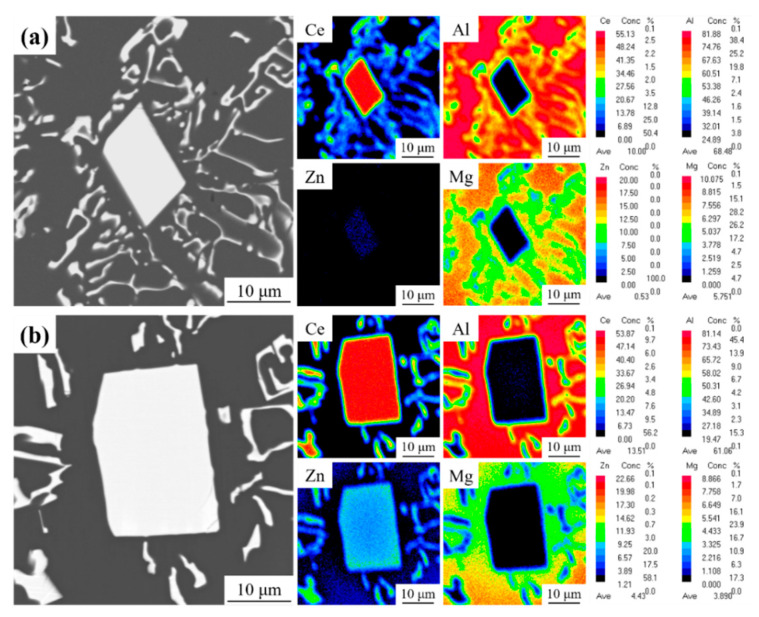
Electron probe microanalysis elemental maps of the as-cast Al-10Ce-3Mg alloys with various Zn concentrations: (**a**) alloy 1 and (**b**) alloy 4.

**Figure 5 materials-17-03999-f005:**
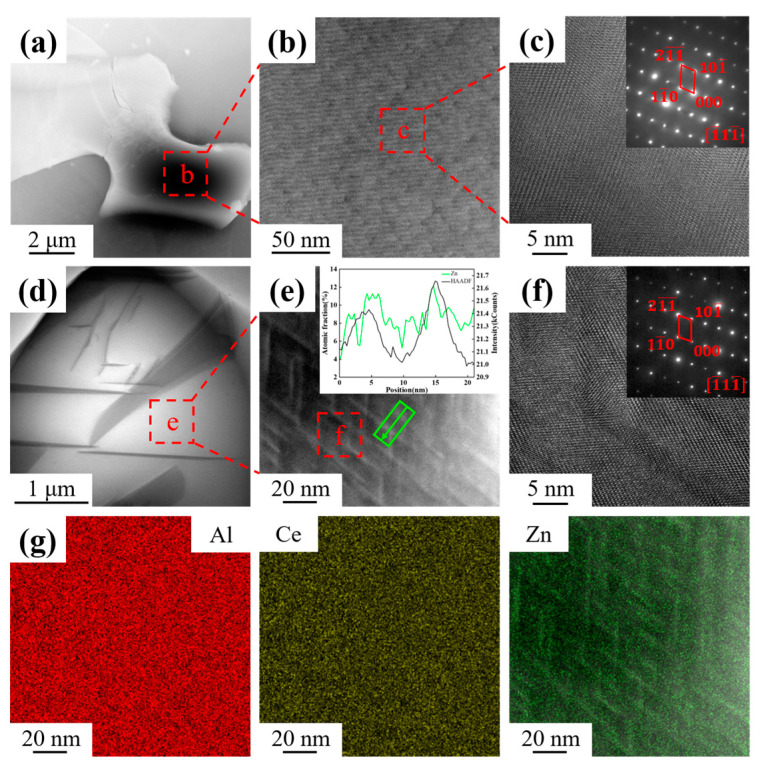
TEM investigations of alloy 1: (**a**) HAADF-STEM micrograph of Al_11_Ce_3_ phase; (**b**) HAADF-STEM image of the square area in (**a**); (**c**) HRTEM image and corresponding SAED patterns of the square area in (**b**); TEM investigations of alloy 4: (**d**) bright-field image of Al_11_Ce_3_ phase; (**e**) HAADF-STEM image of the square area in (**d**); (**f**) HRTEM image and SAED patterns of the square area in (**e**); (**g**) corresponding EDS element maps of (**e**).

**Figure 6 materials-17-03999-f006:**
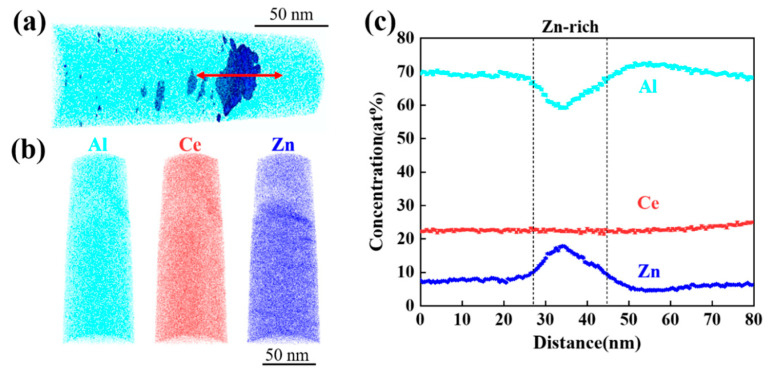
A 3D-APT analysis of the as-cast Al-10Ce-3Mg-5Zn: (**a**) 3-D reconstruction with individual atoms shown, and 20 at% Zn isoconcentration surfaces are delineated in blue; (**b**) 3-D atom maps showing the distribution of each individual atomic species; (**c**) a proximity histogram showing the elemental distributions across a Zn-rich precipitate interface, as indicated by the red arrow in (**a**).

**Figure 7 materials-17-03999-f007:**
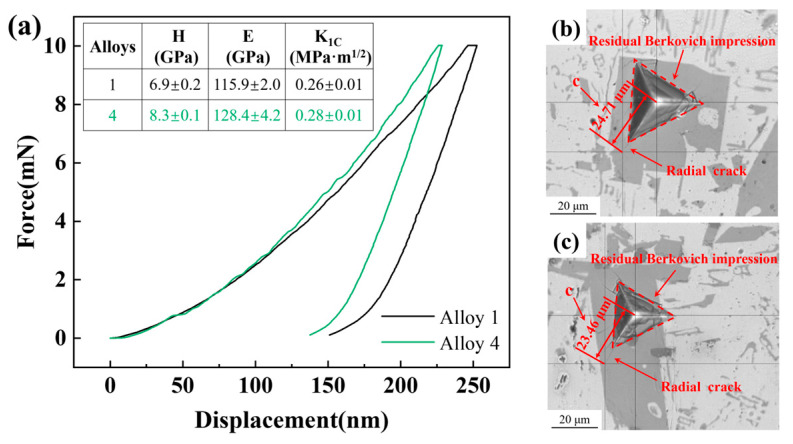
Nanoindentation curves (**a**) and microhardness indents in the primary phase (**b**,**c**) of the as-cast Al-10Ce-3Mg alloys with various Zn concentrations.

**Figure 8 materials-17-03999-f008:**
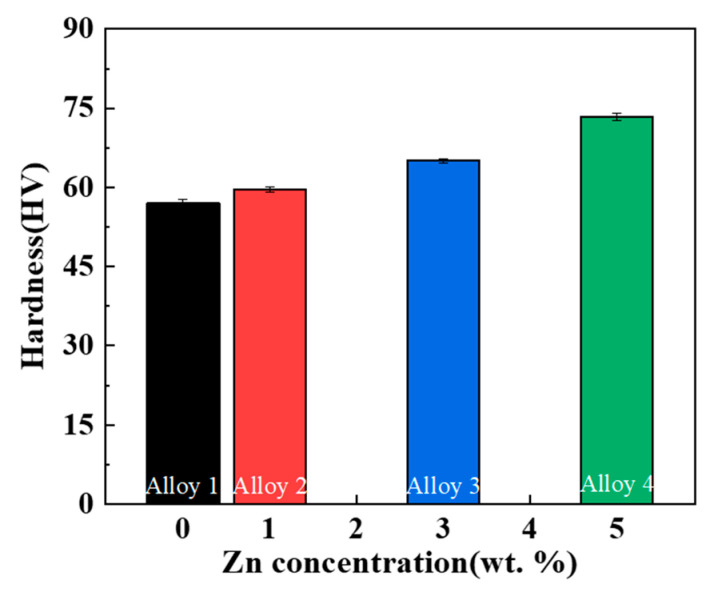
The hardness values of the as-cast Al-10Ce-3Mg alloys with various Zn concentrations.

**Figure 9 materials-17-03999-f009:**
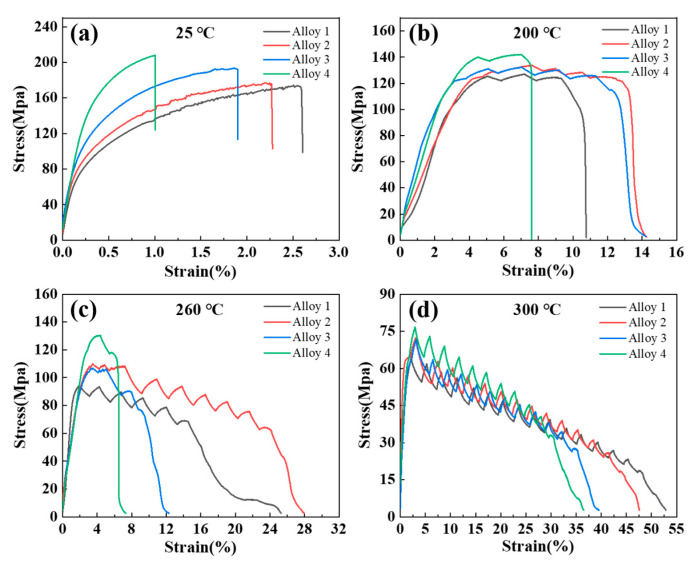
The engineering stress–strain curves of the as-cast Al-10Ce-3Mg alloys with various Zn concentrations at different temperatures: (**a**) 25 °C; (**b**) 200 °C; (**c**) 260 °C; (**d**) 300 °C.

**Figure 10 materials-17-03999-f010:**
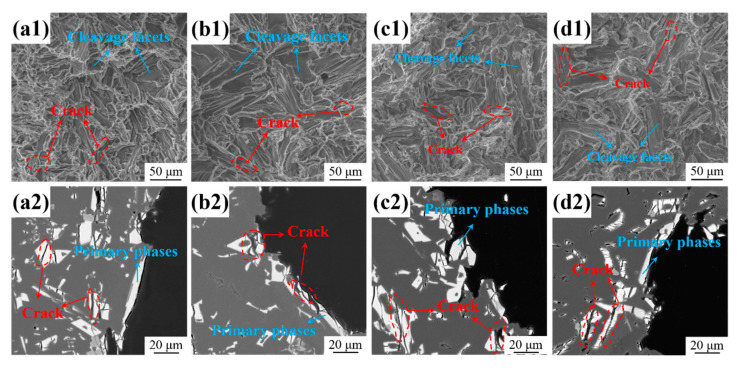
Fracture morphology and the longitudinal sectional area near the fracture surface of the as-cast Al-10Ce-3Mg alloys with various Zn concentrations: (**a1**,**a2**) alloy 1, (**b1**,**b2**) alloy 2, (**c1**,**c2**) alloy 3, and (**d1**,**d2**) alloy 4.

**Table 1 materials-17-03999-t001:** Actual chemical compositions (wt.%) of Al-10Ce-3Mg-xZn alloys determined by ICP-AES.

Alloys	Ce	Mg	Zr	Y	Zn	Al
Alloy 1	10.0	2.8	0.1	0.1	0.0	Balance
Alloy 2	10.5	2.9	0.1	0.1	1.0	Balance
Alloy 3	10.1	3.0	0.1	0.1	3.0	Balance
Alloy 4	10.3	2.8	0.1	0.1	4.9	Balance

**Table 2 materials-17-03999-t002:** EDS results for the points illustrated in Figure 3 (at%).

Point	Al	Ce	Mg	Zn	Phase
1	66.4	33.6	-	-	Al_11_Ce_3_
2	61.4	34.6	-	4.0	Al_11−x_Ce_3_Zn_x_
3	58.6	32.8	-	8.6	Al_11−x_Ce_3_Zn_x_
4	54.2	33.3	-	12.5	Al_2_CeZn_2_
5	96.4	-	3.6	-	α-Al
6	96.4	-	3.3	0.3	α-Al
7	96.2	-	3.1	0.7	α-Al
8	94.8	-	3.6	1.6	α-Al
9	80.3	17.6	2	-	AlCeMg
10	79.3	14.7	2.9	3.1	AlCeMgZn
11	78.1	15.1	2.2	4.6	AlCeMgZn
12	77.5	13.4	2.2	6.9	AlCeMgZn

## Data Availability

The raw/processed data required to reproduce these findings cannot be shared at this time as the data also form part of an ongoing study.

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
