# Peer review of "Microstructure and Mechanical Properties of As-Cast Al-10Ce-3Mg-xZn Alloys"

_materials, 2024, doi:10.3390/ma17163999_

Round 1

Reviewer 1 Report

Comments and Suggestions for Authors

The present study systematically investigated the effects of Zn addition on microstructural evolution and mechanical properties in Al-10Ce-3Mg-xZn alloys.  The manuscript is quite well-constructed with clear English.  However, some parts should be revised for the publication to Materials.  Here, the reviewer would like to give questions and proposals on the current manuscript.

-Page 4-

Line 144: 1C in fracture toughness (K1C) should be expressed in subscript.

-Page 5-

Lines 193-195: The expression “Zn is found to enter into the Al11Ce3” is somewhat confusing because intermetallic compound Al11Ce3 would be composed of only Al and Ce.  The reviewer would like to change the expression not to give rise to such confusions.

-Page 6-

Table 1: For example, the chemical composition at point 1 consists of 66.4 at.% Al and 33.6 at.% Ce (around 3:2) was judged to Al11Ce3.  Could you explain this logical flow on the judgement?

-Page 9-

Lines 267-270

This one sentence used “which” two times, which should be changed.

-Page 10-

Figure 8: I could not find the explanation for Fig. 8 anywhere.  If so, this may be quite critical problem.  Additional explanation should be added in the appropriate place of manuscript.

Lines 303-304: The author interpreted the stress fluctuation in S-S curve as serration.  The serrated flows observed in Alloys 3 and 4 at room temperature (Fig. 9(a)) seem to be serration.  However, some strange fluctuations with very large interval in strain and drastic increases in stress would not be serration phenomenon usually understood in terms of DSA effect.  The reviewer does not agree with this explanation.  Further survey on serration behaviors (Type A, B and C in association with PLC band propagation) is required, and finally addressed for reason of large fluctuations.

Figure 9: The colors of S-S curves are not distinct each other in terms of readability.  There are room for improvement.

-Page 11-

Lines 323-325: The author addressed that the cracks are attributed to different thermal expansion coefficients during solidifications.  Does the author have the evidence to support that? Does the author find the obvious cracks before deformation?

-Page 12-

Overall Section 4.2: The estimation of stress contribution for each strengthening mechanism has been attempted for long time.  The reviewer does not deny such approaches, but the discussion does not involve comparison on the change of Zn additions in both dispersion strengthening and SS strengthening.  Therefore, the discussion looks to me empty.  It should be improved furthermore for more valuable discussion, which would be critical in judgement from reviewer. 

Line 389: What is the scientific meaning of “unrelaxed plastic strain”? Does this indicate total strain (elastic strain + plastic strain)?      

Lines 410-412: Does the author have evidence of entanglement of dislocation? If so, where it happens? Matrix near the phase boundaries?

Equation (6): The shape of Eq. (6) is different from the previous paper the author cited [44].  Please address the difference based on theoretical approach. 

Lines 424-425: How did you determine the Kj=2.9 MPa/wt.%?

Comments on the Quality of English Language

The quality of English is quite clear.

Reviewer 2 Report

Comments and Suggestions for Authors

Dear Authors,

The reviewed article titled: Microstructure and mechanical properties of as-cast Al-10Ce-3Mg-xZn alloys (Manuscript ID: Manuscript ID: materials-3117913) is interesting. The work is well-written. I only have a few questions, which I list below:

Figure 1. X-ray patterns: The authors write that: "The intensity of the Al11Ce3 peak increases with the addition of Zn, indicating an increase in the volume of the Al11Ce3 phase with higher Zn content." This statement is somewhat disputable, as it is clear from Fig. 1 that for Alloy 3, the peaks of the Al11Ce3 phase are less intense than those for Alloy 2, which has a lower Zn content. Please comment on this matter.

Figure 1: Please include in the figure caption an explanation of what the red dashed line signifies, as indicated in the figure.

How does Zn content affect the morphology of the individual phases present in the studied alloys? Does it influence the phase size? If so, how?

High-temperature mechanical properties investigations: The authors explain that the serration behavior of the stress-strain curves is primarily attributed to the interaction between solute atoms and mobile dislocations at elevated temperatures. Can it be said in this case that a dynamic recrystallization process is occurring?

Kind regards

Reviewer 3 Report

Comments and Suggestions for Authors

This study investigates the microstructure and mechanical properties of Al-Ce alloys with the addition of Zn. Extensive tests and analyses have confirmed that adding Zn enhances the mechanical properties of Al-Ce alloys, particularly their strength.

Al-Ce alloy is identified as a promising material for lightweight construction. Enhancing the performance of Al-Ce alloys and broadening their applications can increase the supply and price of Ce, thereby helping to alleviate the imbalance in the use of rare earth elements. The wider application of lightweight aluminum alloys also significantly contributes to energy conservation and environmental improvement.

The problem and the current state of research are clearly described, and the literature references are extensive. The research methodology and results are presented clearly and comprehensively, with high-quality, self-explanatory graphics.

Reviewer 4 Report

Comments and Suggestions for Authors

This work presents interesting mechanical properties of an aluminum alloy compound, mainly studying the effect of incorporating Zn into an AlCeMg mixture. Many techniques have been used to characterize the materials presented in the text (XRS, SEM, TEM, EDS, ATP, nanoindentation,..).
I missed a reference sample prepared without the presence of Zn. That sample would be essential to understand the XRD data (see Figure 1). The nucleation of the Zn phase is sustained by the presence of a strong peak at around 18 degrees. I am surprised that this is the only peak from this compound observed for alloys number 3, 2, and 1. Usually, the peaks with higher intensity in polycrystalline samples are between 30 and 50 degrees. The presence of Zn is minimal (0.04 wt%) in alloy 1, but still, there is a clear peak around 18 degrees. Only peaks between 30 and 50 degrees are observed for alloy 4.
No crystal information is provided for the Al2CeZn2 nor the Al11Ce3 phases, such as crystal structure and lattice parameters, which are fundamental for understanding XRD and TEM data.

Another issue to be consider by the authors is that Kc1 for alloy 1 is 0.26, and for alloy 2 is 0.28. Considering the error, in my opinion, no specific comment can be made about the fracture toughness of the alloys.

Overall, The lack of reference sample (without Zn) in the investigation precluded me from providing a positive recommendation for the publication of this work in Materials

Reviewer 5 Report

Comments and Suggestions for Authors

R E V I E V

of the paper:” Microstructure and Mechanical properties of as-cast Al-10Ce-3Mg-xZn alloys”

ID – 3117913

Line 31 – "...highest temperatures..." is too vague, because it refers to any possible limits. For the work, it is advisable to specify a range of temperatures in which the alloys usually work as you do at Line 38.

Line 32 – The symbols 2xxxx sin 7xxx mean nothing to a non-specialist. It would be good to be more specific because other people can also read the article.

Line 49 - In a scientific work "...superior mechanical properties..." is not indicated to be used. You must either give a comparison of 2 values or indicate with percentages how much higher. As you did at Line. 72.

Line 115 – You indicated the method but did not specify the device used.

Table 1 – The table must have a logic of quantities. Even the studied element does not present this logic. The increase in the amount of Zn is:

- From 0.04 to 1.00, i.e. 25 times.

- From 1.00 to 3.20, i.e. 3.2 times.

- From 3.20 to 4.90, i.e. 1.54 times.

In addition, in alloy 4, you returned to the Ce content of 10.50 without any justification. These questions are generated by the fact that you said that you designed the chemical composition of the alloy, and design means logic in thinking, not random data. Your logic in establishing the quantities from Tab.1 must be explained.

Lines 270 – 276 – Research should not be limited to alloy 1 and alloy 4. Why skip alloy 2 and alloy 3? This work is currently done in the work. The research of the neglected alloys with the same criteria applied to alloys 1 and 4 would have completed the picture of the microhardness variation by varying the Zn content. Jumping from one Zn value to another is not variation.

Line 279 – The same observation regarding "...fracture toughness...". It is stated that in alloy 4 it is higher. Ok! Bigger than who? Only compared to alloy 1? Then why were alloys 2 and 3 developed?

Fig. 8 – There is no comment in the text. Why?

Lines 284 – 286 – It is said that by increasing the Zn content, the UTS increases. If we look at the Zn content of alloys 1 and 2, in the latter the Zn content increases 25 times compared to alloy 1, but the UTS increase is lower than that between alloys 3 and 4 where Zn increases only by 1.54 times. So the simplistic statement of the UTS increase with the Zn content increase (linear statement) should not be made, but this non-linear increase should be explained based on the metallographic evolution of the compounds formed.

Discussion – I think you confused "State of the Art" with "Discussion". In a paper, the "Discussion" chapter must contain comments on the results obtained in one's research. Or you comment on other people's results. It seems to me a mistake in approach that you have to fix.

The calculation relations (2,3,4,5) given in the text had to be presented in "State of the Art" and in "Discussion" they had to be concretely applied to the studied alloys, highlighting what these new alloys brought. If you do not do so, I recommend you move the relationships we refer to in the "State of the Art" of the article.

Line 402 – Again, refer only to alloys 1 and 4 and ignore the comments about the other two. Please fix everything in the text where you exemplify only alloys 1 and 4, commenting on all four elaborated alloys.

Line 425 – Why do you only calculate for alloy 4? I am curious about the values ​​of the other 3 elaborated alloys.

References – I recommend you adopt a writing of the bibliography recommended by the journal because:

- It does not put ”;” between authors.

- The year of publication is placed in parentheses immediately after the authors and in bold.

- The DOI classification for each article is recommended to be done.

- The pages of the article in the volume are marked with pp.... (eg pp.365–385).

- The title of the cited work is placed between quotation marks and written in italics.

- The year of publication is not passed twice (e.g. the work [17]).

General observation

The work is interesting, it caused a lot of work, and it really serves specialists in the field.

However, the work contains many inaccuracies, it has an uneven treatment of the subject, a distribution of citations in the original part of the article, etc.

I recommend correcting all THESE INADVERTENCES IF YOU WANT ACCEPTANCE FOR PUBLICATION.

Round 2

Reviewer 1 Report

Comments and Suggestions for Authors

In the line 288 in the revised vision, "hard-ness" should be revised to hardness.

In overall, revised version would be enough for the publication in Materials. 

Reviewer 4 Report

Comments and Suggestions for Authors

The objections I raised in my report have been addressed and resolved in the new version of this work.
Overall, I recommend the publication of this work in Materials.

Reviewer 5 Report

Comments and Suggestions for Authors

Congratulations for the effort to do major revisions